# Hospital Effluents and Wastewater Treatment Plants: A Source of Oxytetracycline and Antimicrobial-Resistant Bacteria in Seafood

Bozena McCarthy [1], Samuel Obeng Apori [1], Michelle Giltrap [1,2], Abhijnan Bhat [1,3], James Curtin [1,4] and Furong Tian [1,3,*]

1 School of Food Science and Environmental Health, College of Sciences and Health, Technological University Dublin, D07 ADY7 Dublin, Ireland; C17731295@mytudublin.ie (B.M.); apori.samuel@umu.ac.ug (S.O.A.); Michelle.Giltrap@TUDublin.ie (M.G.); abhijnan.bhat@gmail.com (A.B.); james.curtin@tudublin.ie (J.C.)
2 Radiation and Environmental Science Centre, FOCAS Research Institute, D08 NF82 Dublin, Ireland
3 Nanolab, FOCAS Research Institute, D08 NF82 Dublin, Ireland
4 College of Engineering and Building Engineering, Technological University Dublin, D01 YH30 7 Dublin, Ireland
* Correspondence: furong.tian@tudublin.ie

**Abstract:** The present study employs a data review on the presence and aggregation of oxytetracycline (OTC) and resistance (AMR) bacteria in wastewater treatment plants (WWTPs), and the distribution of the contaminated effluent with the aid of shallow and deep ocean currents. The study aims to determine the fate of OTC and AMR bacteria in seafood, and demonstrate a relationship between AMR levels and human health. This review includes (1) OTC, (2) AMR bacteria, (3) heavy metals in aquatic environments, and their relationship. Few publications describe OCT in surface waters. Although OTC and other tetracyclines were found in 10 countries in relatively low concentrations, the continuous water mass movement poses a contamination risk for mariculture and aquaculture. There are 10 locations showing AMR bacteria in treated and untreated hospital effluent. Special effort was made to define the geography distribution of OTC, AMR bacteria, and heavy metals detected in WWTPs to show the likely dissemination in an aquatic environment. The presence of OTC in surface waters in Asia, USA, and Europe can potentially impact seafood globally with the aid of ocean currents. Moreover, low concentrations of heavy metals exert environmental pressure and contribute to AMR dissemination. Recommended solutions are (1) quantitative analysis of OTC, heavy metals, and AMR bacteria to define their main sources; (2) employing effective technologies in urban and industrial wastewater treatment; and (3) selecting appropriate modelling from Global Ocean Observing System to predict the OTC, heavy metals, and AMR bacteria distribution.

**Keywords:** AMR bacteria; *E. coli*; metals; oxytetracycline; wastewater; seafood; human health

## 1. Introduction

Hospital effluents have been perceived to be detrimental to both humans and the environment owing to their hazardous constituents. These effluents consist of pharmaceuticals and their metabolites, diagnostic agents, disinfectants, and pathogens resulting from diagnostic products of laboratories, research activities, and medicine excretion from patients [1,2]. According to Emmanuel et al. (2004), the effluents are metabolized and non-metabolized, either stable or unstable, flowing into the urban wastewater network without appropriate treatments [3]. Waste water treatment plants (WWTPs) ensure high-quality wastewater effluents from the hospitals or industrials into the receiving water bodies, e.g., lagoon, reported by Okoh et al. (2007) and Langford and Thomas (2009) [4,5]. However, WWTPs are not explicitly designed to treat the pollutants presenting in hospital effluents, as hospital effluents are known to be over 150 times more concentrated in micropollutants

than urban or industrial effluents [6]. Meanwhile, in many countries, urban and industrial wastewater is discharged into receiving water bodies without any treatment or after primary treatment only. With this wastewater originating from the hospital, industrial and urban effluents end up in the marine environments through surface runoff and sediment transportation, contributing to the tremendous increase in marine pollution [4].

Marine pollutants, such as antimicrobial-resistant (AMR) bacteria, antibiotics, metals, and contaminants of emerging concern (CECs) are mainly derived from effluents of wastewater treatment plants, hospitals and other healthcare settings (pharmaceutical), industries, and agriculture [7]. In addition, these marine pollutants can affect human health and the food chain through seafood contamination [7]. AMR is a natural and ancient phenomenon resulting from the therapeutic use of antimicrobials in humans and the animal sector [8]. The continuous use of antimicrobials in various applications, including human and animal disease therapies, food animal production, and horticulture, has resulted in high AMR levels in hospitals, urban, and industrial effluents, as well as in wastewater treatment plants [9,10]. These contaminated effluents enter the ocean, finding their way into seafood, and can affect human health when the contaminated seafood is consumed. It has been estimated that, by the year 2050, without new antibiotics, 10 million deaths are likely to occur as a result of AMR. This number is higher than predicted future cancer mortality. The risk of AMR infections will likely prevent many of the medical procedures that we currently take for granted such as routine surgery, chemotherapy, and others [11,12].

Tetracyclines represent one of the major antibiotic families used in human and animal therapy to treat infections and promote growth in livestock. In the medical field, tetracyclines are still perceived as an effective antibiotic against several multidrug-resistant pathogens such as *Staphylococcus aureus*, *Acinetobacter baumannii*, *Klebsiella pneumonia*, and *Escherichia coli*, resulting in its high utilisation in the health and veterinary sectors [13]. However, tetracyclines have been difficult to remove from WWTPs [14]. As a result, agricultural and industrial runoff, as well as treated hospital effluent containing tetracyclines antibiotics, are released into the surface waters, gaining access to the marine environment. Moreover, Ayandiran et al. (2018) found high resistance to tetracyclines in bacteria isolated from poultry farms in Ibadan, Nigeria [15]. Therefore, tetracyclines' family resistance appears to be problematic to the land and water environment stability.

Economou and Gousia (2015) conducted a study on agriculture and food animals as a source of AMR bacteria [16]. Their findings indicated that the AMR bacteria level is increasing in the environment, causing major public health problems. They further attributed the high numbers of AMR bacteria in the environment to the consistent use of antibiotics in farm animals. However, limited studies exist concerning the tetracyclines family resistance in seafood, OTC in particular. Therefore, this study seeks to review the hospital effluents and wastewaters treatment plants as a source of OTC and AMR bacteria in seafood.

### 1.1. OTC and AMR Bacteria

Dissemination of OTC in aquatic systems can be caused by many factors, including inadequately treated sewage, WWTPs, heavy rainfall water discharge, agricultural activities, discharge from trawlers, wildlife, and domestic animal waste ending up in sea waters [17]. *E. coli* resides in human and animal gut as the largest member of the *Enterobacteriaceae* family. *E. coli* are mainly a commensal microorganism residing in the large intestine. However, pathogenic strains can cause gastrointestinal and urinary tract infections, meningitis, and sepsis. Exposure usually occurs through contaminated water, food, and contact with animals [18].

The use of tetracyclines considerably increases the prevalence of bacterial strains, including *E. coli* [19]. Moreover, antibiotics are released into the surface aquatic environment in biologically active form, affecting bacteria in their natural habitat. Heavy rainfall also causes an increased amount of sewage and animal manure runoff into the coastal

waters. Thus, a mixture of bacteria, organic substances, and antibiotics creates an excellent environment for horizontal gene transfer (HGT) and AMR dissemination.

It has been reported that hospital wastewaters (HWWs) are a hotspot for AMR bacteria and AMR genes [18,20]. Furthermore, WWTP effluent is released into water bodies such as rivers, canals, and coastal seawaters [21]. Moreover, hospital effluent may be utilized to monitor and evaluate AMR resistance genes in bacteria and humans [22]. Notably, developing countries with large populations appear to heavily experience the occurrence of AMR [23]. Blaak and colleagues (2015) proposed that manure runoff from livestock production leads to the presence of faecal bacteria such as *E. coli* in surface waters. Moreover, when the agricultural effluents are drained into rivers, ponds, and eventually seas, *E. coli* may disseminate easily into the aquatic environment [24].

The tetracyclines family are broad-spectrum antimicrobials that include a vast group of compounds considered as safe. These antibiotics are commonly prescribed treatments for bacterial infections in humans and livestock globally. Consequently, a steady increase in AMR in healthcare settings and populations across the globe has been observed. Tetracyclines consist of several main groups, including tetracycline (TC), OTC, tetracycline (TC), demeclocycline, lymecycline, doxycycline, minocycline, and tigecycline (TIG) [25]. All of these act on bacteria protein synthesis responsible for reproduction. Once antibiotics access the cell, they compete with transfer RNA (tRNA) for the large ribosome binding site. Once bound, tetracyclines elicit a series of events that eventually cause altered chromosome structure. Thus, tetracyclines successfully inhibit bacterial replication by inactivation of protein synthesis. However, gram-positive and gram-negative bacteria exhibited resistance by prevention of accumulation of tetracyclines through antibiotic efflux [26].

Other pathways that give rise to AMR in bacteria also exist and pose a threat. AMR to penicillins and cephalosporins was also reported in hospital effluent in eastern France [27]. Certain bacteria can produce β-lactamases enzymes that inactivate antibiotics by altering their structure. In addition, genetic material expresses AMR genes that produce proteins to protect itself from antimicrobial action. The semi-permeable bacterial cell membrane allows β-lactams antibiotics to access the cell via porins, where they bind to penicillin-binding proteins (PBPs). This process leads to one or a few events such as enzymatic inhibition of antibiotics, decreased membrane permeability, and active efflux of antibiotics through the efflux pump mechanism [18].

### 1.2. Techniques for Treatment of HWW

Antibiotic-resistant genes (ARGs) and pharmaceutically active compounds (PhACs) in various wastewater treatment processes are classified into four distinct categories (physical, chemical, physiochemical, and biological methods) [28]. The main technologies for treatments of wastewater are described. A comparison and advantages of novel wastewater treatments methods are also explored.

#### 1.2.1. Chemical Process

Chemical techniques used for treating HWW include catalytic ozonation and chlorination. The catalytic ozonation process is used to remove AMR bacteria and ARGs from HWW. This is achieved through deactivation of pathogens in wastewater via direct reaction with ozone, and by the indirect action of hydroxyl radicals [29]. The effectiveness of the catalytic ozonation process depends on ozone concentration, pH value, catalysts' dosage, and catalyst stability. Moreover, AMR bacteria species, reaction time, and concentration of microbial communities can also influence the effectiveness of ozonation [30]. Oh et al. (2014) examined the effect of ozonation on the removal of AMR bacteria. High 90% removal efficiency of the AMR bacteria (pB10 Plasmid DNA) was found. This can be attributed to degradation of the structural integrity of the ARB cell envelope and results in the deactivation of cellular constituents by the ozone. However, the catalytic ozonation is limited to in situ remediation of HWW [31].

The chlorination process employs a similar mechanism of action to ozonation to remove ARBs, ARGs, and PhACs in HWW. The removal efficiency of ARGs and AMR bacteria in HWW depends on chlorine dosage, reaction time, microbial concentration and phenotype, ARGs' category, and overall water quality [32]. Furukawa et al. (2017) obtained a 3.59 log reduction of vanA (common in healthcare settings) gene via wastewater treatment with 3.0 mg $Cl_2$/L for 60 min. [33]. However, Jia et al. (2019) observed that ARGs increased after the chlorination process [34]. Moreover, a similar observation was revealed by Destiani et al. (2019), and it was concluded that chlorination stimulates the regrowth potential of ARB [35].

### 1.2.2. Biological Process

Conventional biological technologies such as activated sludge process (ASP), membrane bioreactor (MBR), and constructed wetlands (CW) are widely used for the treatment of HWW, as reported in Majumder et al. (2020) [36]. However, the effectiveness of the biological process for remediation of PhACs and ARGs in HWW depends on the complex structure of the PhACs. In addition, the type of process and ARGs can also impact the biological methods [37,38]. Some of the ARGs and AMR bacteria that have been reported to be remediated by MBR technology are as follows: tet(W), tet(O), sul1 tetC, tetE, sul1, sul2 tetC, blaCTX-M-1, and blaCTX-M-9. Despite MBR being a powerful technique, membrane fouling and repeated washing are the main factors that can limit the application of MBR on a large scale. In comparison, the CW method showed more promising results on a large scale. However, the effectiveness of MBR for removal of PhACs, AMR bacteria, and ARGs in HWW depends on microbial community, antibiotic loading, and types of ARG and PhACs present [39–41].

In addition, conventional HWW treatment technologies (ASP, MBR, and CW) are ineffective in total removal of antibiotics and other pharmaceuticals. In Spain, only 50% of the antibiotic load was removed [42]. Oflaxin removal rates in East China were between 44.2 and 81.6% Moreover, as little as 36.2% of ampicillin was still present in treated hospital effluent [43].

Research into novel techniques to eliminate AMR bacteria and ARG genes has picked up pace in recent years. 99.00% of *S. aureus* was removed by enhanced filtration using oxidized cellulose, and coupling of tannic acid and gelatine polymer. An improvement in conventional adsorption was achieved using magnetic graphite oxide technique. This treatment presents antimicrobial effectiveness up to 98.97, 97.15, and 97.67% for *E. coli*, *Yersinia ruckeri*, and *Enterobacter agglomerans*, respectively [44].

### 1.2.3. Physical Process

Physical technologies such as coagulation, membrane separation, and the adsorption process for treatment of HWW are widely used. The coagulation process uses chemical coagulants (ferric-based coagulants and aluminium-based coagulants) for the removal of AMR bacteria, ARGs, and PhACs. Grehs et al. (2019) observed high removal of blaTEM and qnrS using aluminium-based coagulants. High eradication efficiency was attributed to high doses of coagulants used [45].

The adsorption process is described as very effective in conventional water treatment, and inexpensive owing to the low cost of adsorbents. Activated carbon, silica, clay, and metal oxides are extensively used as adsorbents because of their high adsorption capacity [46]. For example, magnetic biochar has displayed effective removal of *E. coli* and *S. aureus*, as reported by Fu et al. (2020) [47].

Membrane separation technology uses physical separation techniques such as nanofiltration, reverse osmosis, microfiltration, and ultrafiltration. All of these create a physical barrier to effectively remove pathogens from HWW, thereby reducing the abundance of the AMR bacteria and ARGs entering into the environment. Hospital effluent is subjected to an initial filtration process with the use of trickling biofilters equipped with stones or other materials able to trap the microorganisms. Upon the attachment, organic matter present

is diffused and broken down. The liquid achieved after filtration is subjected to further processing [46].

### 1.2.4. Physiochemical Process

Additionally, disinfection is used to eradicate AMR bacteria and ARGs in HWW. Disinfection can be achieved by application of UV irradiation, resulting in irreparable oxidative damage to bacterial DNA. However, the light irradiation can also trigger an oxidative stress response or alter gene expression. Upregulated gene expression allows bacteria to resist the damaging effects of toxic agents, especially when first pre-exposed to UV light stimulus [48,49]. Therefore, more studies are needed to provide a deductive mechanism of UV irradiation used for HWW treatment.

## 2. Materials and Methods

### 2.1. Search Strategy

The systematic search and review processes were conducted following the Preferred Reporting Items for Systematic Reviews and Meta-Analyses (PRISMA) Statement criteria reported by [50]. For the present study, research articles were searched on Google Scholar (https://scholar.google.com/, accessed on 25 November 2021) and Scopus databases (https://www.scopus.com/home.uri, accessed on 25 November 2021) using the following search terms: "AMR and OTC" successfully combined with "hospital treated and untreated wastewater", "seafood", "human health", "hospital effluents", "wastewater treatment plant", and "seafood contamination". Scopus and Google Scholar were used for the review owing to their largest dynamic reference information base explored for writings incorporating logical diaries, books, and gathering procedures [51]. To further ensure that we had assembled a comprehensive list of studies, we asked researchers having the relevant knowledge on the topic to review and suggest additions to the keywords. The search was limited to scientific articles published between 2010 and the current date of conducting the review (2021), and yielded 100 research papers.

The literature search was limited to the following:

- Articles' publication years were between 2010 and 2021;
- The keywords AMR and OTC successfully combined with "hospital treated and untreated wastewater", "seafood", "human health", "hospital effluents", "wastewater treatment plant," and "seafood contamination" in the title and abstract;
- The articles had to be scientific indexed papers only;
- Search was limited to research articles only.

The retrieved articles were imported using Zotero 2.03 and duplicate records were deleted and scrutinized. The results were screened against inclusion criteria, i.e., articles that are not relevant to the studies. Full text of papers for all the articles that fitted into the inclusion criteria was retrieved. Articles were being excluded for the following reasons:

- They were published in languages other than English;
- Articles that only an abstract were available;
- Articles that are not related to the studies were also excluded.

### 2.2. Data Extraction and Reporting

A standard, purpose-designed form was adopted and modified from Armah et al. (2014) for extraction of the following data from the paper [52]:

- Location, sample size (L), total samples, WWTP effluent, WWTP influent, and AMR genes detected;
- Types of tetracycline antibiotic and region of the study;
- Results, including a mean antibiotic concentration in hospital effluents and wastewater treatment plants.

Most of the articles included in this review (>90%) provided measures of central tendency, that is, arithmetic measures, and a few of them were usually accompanied by

standard deviations (SDs). In this study, AMR *E. coli* data extracted from the papers was limited to cfu/mL values, and favoured over percentage to highlight precisely the bacteria cell count and distribution and construct bar chart (Figure 1). Therefore, only eight publications were related to AMR as a result of hospital WWTPs. Variance in sample size (between 0.05 and 1 L), the total number of samples (1–48), hospital effluent samples (0–24), and WWTP influent samples (between 0 and 24) was included in the analysis at the same unit.

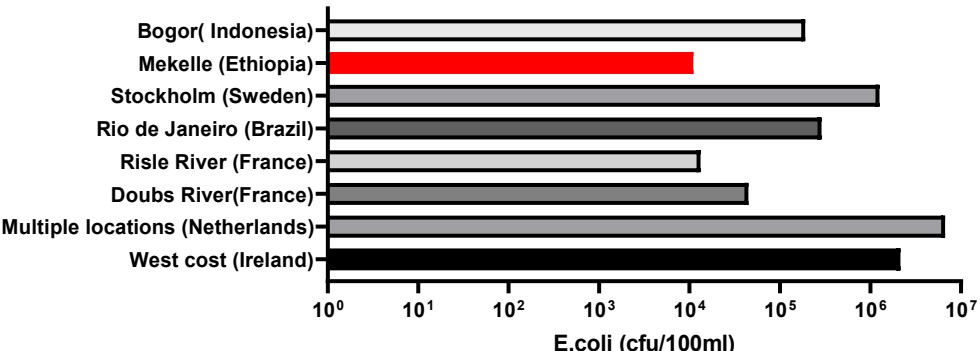

**Figure 1.** AMR *E. coli* levels in treated and untreated hospital effluent in different regions around the world. In black: untreated hospital effluent; in red: treated hospital effluent.

## 3. Results and Discussion

### 3.1. Levels of OTC and AMR Bacteria in HWW

Ten papers reported on research related to tetracyclines in WWTPS, as presented in Table 1. Only two papers showed tetracyclines and heavy metals in WWTPs (*n* = 2). The average results of *E. coli* numbers are presented in Section 3.2. AMR genes in bacterial genomes (*n* = 8). AMR concentrations are displayed in a bar chart, as listed below, while the locations of tetracyclines, heavy metals, and AMR bacterial are shown by green, yellow, and red on the map, respectively, in Section 3.6. OTC and AMR distribution globally.

**Table 1.** Environmental concentration of antibiotics of the tetracycline family.

| Antibiotic | Mean Antibiotic Concentration (ng/L) | AMR Bacteria or ARG | Matrix | Region | Reference |
|---|---|---|---|---|---|
| TC | N/D | Resistant *E. coil* 3264 cfu/100 mL | Hospital effluent | West coast, Ireland | [7] |
| TC | 10 | N/D | Hospital effluent | Risle river, Northern France | [27] |
| TC<br>OTC | 1.9<br>3.8 | N/D | WWTP municipal Effluent | Beijing, China | [53] |
| OTC<br>TC | $32.0 \times 10^7$ OTC<br>$2.6 \times 10^6$ TC | N/D | PWWTP Influent and effluent | North China | [54] |
| TIG | N/D | *bla*$_{NDM1}$ | Seepage and tap water | New Deli, India | [35] |
| OTC | N/D | $4.8 \times 10^5$ cfu/100 mL in river<br>$4.8 \times 10^6$ cfu/100 mL in WWTP<br>*tetM* detected in surface water | River, WWTP, and surface water | Coast of North- East South Africa | [55] |
| TC | N/D | *tetM* detected in 100% bacteria in all three locations<br>*tetC* detected in 80% bacteria Helsinki,<br>27% Tallin, and<br>73% Tartu | WWTP effluent | Helsinki, Finland Tallin, Estonia Tartu, Estonia | [56] |
| OTC | 70–1340 ng/L | N/D | Surface water | USA | [57] |
| OTC | up to 340 ng/L | N/D | Surface water | UK | [57] |
| OTC | 71,700 ng/L | N/D | Runoff | England | [57] |

N/D—not determined; OTC—oxytetracycline; TC—tetracycline; TIG—tigecycline; WWTP—wastewater treatment plant; PWWTP—pharmaceutical wastewater treatment plant; ARG—antimicrobial resistance genes.

A bar chart was constructed to illustrate the AMR concentration from different countries (Figure 1).

### 3.2. AMR Genes in Bacterial Genomes

A study conducted in Rio de Janeiro (Brazil) found that *E. coli* was present along with many other AMR bacteria such as *Pseudomonas* spp., *Enterobacter*, and *Klebsiella* in hospital and WWTP treated effluent [58]. Among detected bacteria, resistant *Pseudomonas* spp. has been placed on the World Health Organisation (WHO) list of bacteria for which antibiotics are critically needed [59]. Notably, extended-spectrum β-lactamase (ESBL) producing *Enterobacteriaceae* such as *E. coli*, *Enterobacter*, and *Klebsiella* are also a pathogenic bacterium highly resistant to many antibiotics [60].

The selection of eight papers for data mining is outlined in Table 2; AMR *E. coli* levels are clustered to define the most significant region among the eight regions around the world. All of the samples are obtained from WWTPs, which served at least one nearby hospital. Because of the lack of a standardised sampling method, the following limitations are noted. Owing to the heterogeneous nature of the sample, a varying number of samples drawn at a different time of the day, month, and year are the main variables. All of the studies, except one, present the findings on resistant *E. coli* numbers in untreated hospital effluent. *E. coli* is found to be resistant to one or more antibiotics. AMR susceptibility testing methods found in the study include antimicrobial disk, polymer chain reaction (PCR), pulsed-field gel electrophoresis (PFGE), multilocus sequence typing (MLST), and check-points CT101 microarray. All of the studies except for those in Stockholm and Sweden have employed antimicrobial disk. In addition, the antibiotic resistance indicator bacteria (AREB) test was employed in Stockholm and Sweden. Six papers that exploited additional approaches to gene detection are presented in Table 2; however, one study did not determine sample size.

**Table 2.** Procedures used in *E. coli* detection in hospital effluent and AMR genes detected.

| Location | Sample Size (L) | Total Samples | WWTP Effluent | WWTP Influent | AMR Genes Detected | Reference |
|---|---|---|---|---|---|---|
| West coast, Ireland | 1 | 44 | 17 | 0 | *blaCTX-M (blaCTX-M-28, blaCTX-M-3, blaCTX-M-61, blaCTX-M-15 blaCTX-M-14), blaTEM, blaSHV* | [7] |
| Netherlands (North Sea) | 1 | 5 | 5 | 0 | *blaOXA* | [24] |
| Doubs river, Besancon, Eastern France | N/D | 1 | C | 1 | *blaSHV* | [61] |
| Risle river, Northern France | 1 | 48 [1] | 24 [1] | 24 [1] | *blaTEM* | [27] |
| Rio de Janeiro, Brazil | 1 | 3 | 0 | 8 | | [58] |
| Stockholm, Sweden | 0.05 | 6 | 6 | 0 | *blaCTX-M (blaCTX−M group1, blaCTX−M group9, blaCTX−M group2)* | [62] |
| Mekelle, Ethiopia [2] | 0.125 & 0.25 | 20 | 20 | 0 | *blaSHV,* | [63] |
| Bogor, Indonesia | 0.25 | 1 | 0 | 1 | *blaTEM* | [64] |

[1] mean; [2] treated hospital effluent; N/D—not determined.

Tetracyclines including OTC, TC, and TIG were detected in ten regions. OTC was found in Beijing in China, North China, USA, United Kingdom, and England. Municipal WWTP treated effluent contained 3.8 ng/L of OTC in China [53]. OTC of $32.0 \times 10^7$ ng/L was found in North China in pharmaceutical WWTP (PWWTP) [54]. The findings revealed in South Africa suggested high dissemination of OTC resistance bacteria in the

environment [55]. OTC of $4.8 \times 10^5$ cfu/100 mL resistant bacteria was found in a river near North-East South Africa [55]. WWTP located in the same region of South Africa obtained $4.8 \times 10^6$ cfu/100 mL of bacteria containing resistant genes. Moreover, t*etM* gene associated with OTC resistance was found throughout river water [55]. Surface water in USA had 70–1340 ng/L of OTC. Similarly, in the United Kingdom, up to 340 ng/L and 71,700 ng/L of OTC was found in surface water and runoff, respectively.

TC was detected in six locations: Ireland, France, Finland, Estonia, China, and North China. Hospital effluent on the Irish West coast had 3264 cfu/100 mL of TC resistant *E. coil* [7]. Risle river in Northern France and WWTP municipal effluent in Beijing, China, contained 10 and 1.9 ng/L TC, respectively [27,53]. Significant findings in Helsinki, Finland; Tallin, Estonia; and Tartu, Estonia were obtained. WWTP effluent contained *tetM* gene in 100% of bacteria in all four locations. Further, *tetC* gene was detected in 80, 27, and 73% of bacteria in Helsinki, Tallin, and Tartu, respectively [56]. In PWWTP in North China where the OTC was found, a $2.6 \times 10^6$ cfu/100 mL of TC was present in the effluent [54]. In New Delhi, India, seepage and tap water contained bacteria with $bla_{NDM1}$ present associated with TIG, a member of the tetracyclines antibiotics family [65]. Borgi and Palma (2014) suggested that OT and doxycycline are found in concentrations below the limit of measuring the effects on bacteria and fish and shellfish. This poses a challenge in detecting tetracyclines in aquatic environments and hospital/municipal wastewaters [57].

In total, 20% of *E. coli* isolates from the human intestine was found to be resistant to TC. However, administration of TC for 10 weeks (500–1000 mg/day) resulted in significantly increased resistance, up to 96% [57]. Therefore, it is imperative to establish an OTC connection with AMR bacteria and resistant genes.

Coagulase-negative staphylococci (CoNS) including *Staphylococcus epidermis* are commensal bacteria found on human skin. It is understood that *S. epidermis* is usually not associated with high morbidity in humans [66]. However, it can cause pneumonia in premature infants and post-surgery infections in older kids [59]. In Ethiopia, CoNS isolated from hospital effluent were found to be 100% resistant to penicillin and half of the isolates showed tolerance to Cefoxitin. ESBL-producing *S. aureus* was 100% antibiotic tolerant when found in treated effluent. As much as 77% and 33% of *S. aureus* isolates were resilient to penicillin and cefoxitin, respectively. Similarly, 100% resistance to ampicillin was detected in *Klebsiella* spp. and *Citrobacter* spp. [67]. In Turkey, 40.9% of blood infections in geriatric health care setting were caused by ESBL-producing and carbapenem-resistant *Klebsiella* [68].

In addition, vancomycin-resistant CoNS (VRCoNS) and vancomycin-resistant enterococci (VRE) are examples of pathogens commonly found throughout hospitals and other healthcare facilities. Infections with high morbidity and mortality are associated with VRCoNS and VRE including skin and wound infection, urinary tract infections (UTIs), blood infections, sepsis in infants, and meningitis, among others [69].

Few data exist on recent prevalence of VRE and VRCoNS in HWW. This presents a problem in describing the magnitude of dissemination of these pathogens in HWW, especially in developing countries. However, in Bahir Dar, Ethiopia, CoNS and VRE were found in patients' blood, urine, and wounds, at 12% and 34.61%, respectively [69]. In hospital in Iran, 33.4% prevalence of VRE was also found [70].

Hospital effluent was found to contain *Acinetobacter baumannii* and *Pseudomonas aeruginosa*, both carbapenem-resistant, and vancomycin-resistant *Enterococcus faecium*. The levels of ARGs and antimicrobials were higher when compared with other sources [71].

Once reaching the peak in the bloodstream, pharmaceuticals such as antibiotics are excreted by the kidneys in the form of urine. Antimicrobials may also be eliminated by the liver through excretion into bile and finally removed in faeces. However, urine remains the main path of antibiotic excretion [72].

In summary, besides *E. coli*, there are other AMR bacteria present in healthcare facilities. These bacteria and ARGs may access HWW by excreted urine, other bodily fluids, and faeces [67,73].

### 3.3. Bacterial Resistance to Antimicrobials

ESBL-producing *E. coli* was found in untreated HWW in eight regions worldwide, with the lowest levels observed in Northern France. The highest numbers of resistant *E. coli* were detected in the Netherlands in the North Sea region (Figure 1). The wastewater pathway (hospital–WWTP–river Risle) was evaluated for resistant *E. coli* presence in Risle river in France. Although resistant *E. coli* levels decreased along the hospital–WWTP–river continuum, amoxicillin, ticarcillin, and TC resistant strains remained [27]. Resistant *E. coli* presents in the river even after antibiotics are hydrolysed. River Risle eventually ends up in the English Channel, which serves as a large fishing farm. OTC is one of the most commonly used antibiotics used in fish farms. Moreover, OTC, chlortetracycline (CTC), and TC are difficult to remove in WWTPs [54].

In Rio de Janeiro in Brazil, bacterial population reduced upon high levels of chlorine treatment [58]. However, the number of certain resistant strains was reported to be higher after the treatment owing to the development of resistance genes [58,61]. There are only 3 out of 127 hospitals in this region that have treated wastewater. Tetracyclines were found in the effluent of WWTPs [58]. It is indeed necessary to develop sufficient wastewater treatment in Brazil.

In Bogor, Indonesia, nearly all *E. coli* was amoxicillin and erythromycin-resistant [64]. Similarly, *E. coli* is resistant to penicillin in an aquatic environment in another location in Indonesia (Sumatra) [74]. A similar result was obtained in New Deli in India, where high levels of ARGs associated with resistance to TC, sulfonamide, and β-lactam were detected in all stages in WWTP [65].

In Ethiopia, treated hospital effluent was exceedingly high in resistant *E. coli*. Besides, a study from Eastern Cape, South Africa, concluded that resistant bacteria rates are higher after chlorine treatment [75]. A study in Bangladesh determined tetracyclines' resistance in HWW as a contributing factor in multidrug resistance phenomenon [76]. Recently, Chen et al. [77] investigated the transmission of ARGs between bacteria under a variety of light conditions. Their study findings revealed that antibiotic-resistant strains *E. coli* DH5alpha and *E. coli* C600 have stress responses to simulated sunlight and UV irradiation [77,78]. Many studies have also shown that wastewater treatment with chlorination is ineffective in eradicating resistant bacteria in Ireland, Brazil, France, Poland, Austria, Switzerland, and China [7,58,61,79–84].

Moreover, a study from Eastern Cape, South Africa, illustrates that resistant bacteria rates are higher after chlorine treatment [75]. There are 19 TC resistant genes from wastewater used for urban agriculture in West and South Africa as reservoirs for antibacterial resistance dissemination. Further, more advanced and effective wastewater treatments methods to remove AMR bacteria are desirable [83,85].

### 3.4. Heavy Metals as AMR Genes' Co-Regulators

Heavy metals are among the substances found to fuel AMR dissemination. These compounds are found naturally in the environment. In addition, urbanized areas and agriculture are a source of high heavy metal levels [86,87].

Antibiotic resistance may occur through bacteria mutation and horizontal gene transfer (HGT). HGT is a series of processes, including conjugation, transformation, and transduction. Conjugation is a transfer of genetic material, usually plasmid, which requires two bacterial cells to be close to one another, and it can be intra-species and inter-species events. However, the latter takes place less regularly. The transformation is described as the uptake of unprotected DNA by the recipient bacteria cell. Transduction involves the use of bacteriophage viruses as vesicles to transport genes [87].

HGT may also be promoted by antibiotics and other microbial agents in levels below minimum inhibitory concentration (MIC) [88]. Moreover, heavy metals such as $Cu^{2+}$, $Zn^{2+}$, $Ag^{2+}$, and $Cd^{2+}$ may increase oxidative stress and genotoxicity in bacteria, causing the cell cycle to be disrupted, impairing DNA repair and replication [86]. In addition, heavy metals

can cause bacterial mutation and transfer AMR genes through HGT. For example, *E. coli* was found to engage in the conjugative transmission of AMR genes to other genera [87].

Co-selection is a term describing a collection of multiple antibiotic resistance genes where one gene is expressed. Gene resistance to one antibiotic may be responsible for resistance to multiple antimicrobials. Moreover, bacteria may be resistant to antibiotics and heavy metals. Genes participating in this process are located in a mobile genetic element that can be transferred between bacteria. These include structures such as plasmids, transposons, and integrons. However, a bacterial plasmid containing a certain number of genes, including AMR, was usually the primary genetic structure involved in AMR dissemination across bacterial clusters [86].

Heavy metals affect AMR genes' dissemination in the aquatic environments. When $Cu^{2+}$, $Zn^{2+}$, and $Cd^{2+}$ concentrations are above MIC, lower prevalence of conjugative transfers is observed. However, heavy metals are usually present in the natural environment at levels below MIC. Sub-lethal values of Cu, Zn, Cd, Cr, Pb, Ag, and Hg involve intracellular ROS generation, increasing cell membrane permeability, inducing oxidative stress and the SOS response, and altering gene expression conjugative transfer [86].

### 3.5. Connecting OTC, Heavy Metals, and AMR Bacteria

OTC interactions with $Cu^{2+}$, $Zn^{2+}$, and, $Cd^{2+}$ in the aquatic environment were studied in China. OTC binds heavy metals through electron donation, which poses a significant environmental implication. The antibiotic and heavy metal complex exhibited higher toxicity than OTC and heavy metals alone. Moreover, the OTC metal complex was associated with higher AMR genes in aquatic environments [89].

It was established that OTC and heavy metals such as $Zn^{2+}$ and $Pb^{2+}$ reside in soil. Owing to the continuous movement of the water cycle in the environment, it was suggested that the OTC metal complex can be absorbed by the soil and eventually access rivers and coastal waters [90].

Manila clam (*Ruditapes philippinarum*) was studied to present AMR bacteria and heavy metals in Korea. Here, 42% of *Aeromonas* spp. isolates were resistant to OTC. Moreover, they were identified as a factor in developing *bla*TEM and *qnr*S resistance genes, among many others. Further, $Cu^{2+}$, $Zn^{2+}$, $Cd^{2+}$, and $Cr^{2+}$ were also present in *R. philippinarum* clam [91].

The reuse of water causes higher risk in Asian countries [21]. A high probability of AMR transmission exists in hospitals and the highest average relative abundances of *tetX* genes in China's south region [90,92]. The *tetX* genes resistant to TC came from different sources, including aquaculture and agriculture. The *tet*-resistant gene was necessary to draw an OTC distribution, heavy metal, and AMR *E. coli* globally to understand the bacteria's fate and impact on seafood and human health [93].

### 3.6. OTC and AMR Distribution Globally

Figure 2 represents regions where OTC-resistant *E. coli* and heavy metals were detected. North and South Atlantic currents were also outlined to illustrate how AMR is present in effluent traveling around the Atlantic coasts. For example, AMR and ARGs in Rio de Janeiro effluent will arrive at the Western African coast and, eventually, end up in Central and North America, and then even further in Western Europe [94].

The river Risle mouth is located at the English Channel, which serves as a large fish farm and trawl fishing location. OTC is one of the most used antibiotics in fish farms. According to the U.K. Sea and Fisheries Statistic report (2019), 21%, comparable with 31 thousand tons of shellfish, was sourced from English Channel. Fish such as plaice, sole, cod, and mackerel are also found in the Western and Eastern Channel (insert in Figure 2). Fish farms, including salmon in Brittany (France), are located along the Channel coasts [95,96]. These findings are troubling as tetracyclines in Risle river may access seafood in the Channel and access human microbiota upon ingestion.

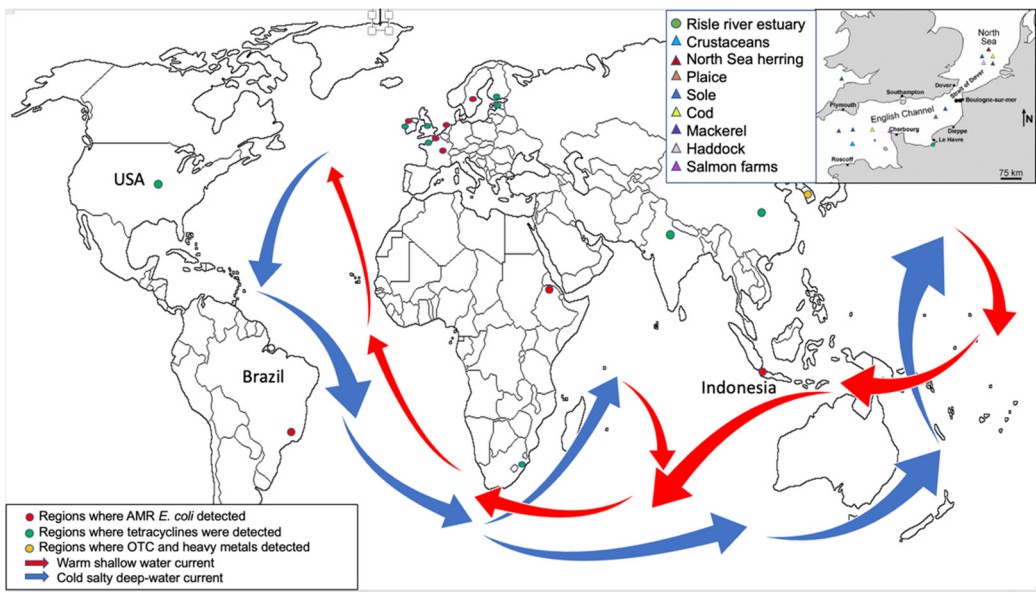

**Figure 2.** Illustration of regions where AMR *E. coli*, tetracyclines, and OTC and heavy metals distribute around the world, and a map of North and South Atlantic currents contributing to AMR spread. Insert shows the English Channel.

Furthermore, it was found that the North Sea serves as a large fishing location for the U.K. population. Species of herring, sole, cod, haddock, and mackerel are sourced in the North Sea region. Tetracyclines' resistance was detected in Denmark in farmed fish and river fish in 29% and 6%, respectively [97]. Therefore, OTC released near North Sea Coasts may enter seafood and pose a risk to human health. Plaice and cod sourced from the Eastern English Channel and French coastal shallow waters were tested for heavy metals. In cod, cadmium, copper, manganese, and lead mean values (from three different locations) were found at 0.07, 9.06, 11.1, and 0.15 µg/g, respectively. In the dry weight of cod muscles, cadmium, copper, manganese, and lead were found at 0.008, 1.23, 1.35, and 0.027 µg/g, respectively [98].

Similarly, in the dry weight of plaice liver, cadmium, copper, manganese, and lead were detected at 0.26, 10.8, 5.7, and 0.24 µg/g, respectively. In the muscle, 0.02, 1.425, 1.29, and 0.05 µg/g of Cd, Cu, Mn, and Pb were observed, respectively. Comparing the above results with a study conducted in 1989, lead and cadmium values detected in the 2004 study were lower [98]. However, as previously mentioned in the present study, low concentrations of heavy metals contribute to AMR dissemination in the aquatic environment.

According to Reverter et al. (2020), warmer months may have influenced bacteria to replicate more efficiently than cold weather. Additionally, many variables such as global warming may have an impact on the AMR bacteria concentration [99]. Finally, the tropical climate on the Africa, Asia, and America continents may have impacted the AMR concentration and colder European countries. Ocean models play a large role in understanding the ocean's influence on weather and climate. Global Ocean Observing System and Integrated Ocean Observing System provide efficacious modelling methods. The optimised selection of modelling from the open accessed Observing System to fit each ocean geography will open a new chapter to predicting OCT, heavy metal, and AMR distribution.

## 4. Conclusions

Hospital and WWTP effluent are indeed a source of antimicrobials and AMR bacteria in the aquatic environment. The use of antimicrobial agents, namely OTC and other tetracyclines, in the treatment and prevention of pathogenic infection in humans and animals may be responsible for the accelerated spread of bacterial resistance in the aquatic environment. The present study highlights the importance of further research needed into



the fate of AMR bacteria in the aquaculture and aquatic environments, and its effects on human health.

The OTC and ESBL-producing *E. coli* was studied in untreated HWW in 18 regions around the world. The OTC were detected in WWTPs in Asia, USA and Europe. The lowest levels of AMR *E. coli* were observed in Northern France. The highest numbers of resistant *E. coli* were detected in Netherlands in the North Sea region.

The research to date contributed to an understanding of OTC and AMR *E. coli* existence and its spread in the aquatic environment. The presence of OTC and low concentration of heavy metals have an effect on development of AMR genes. The OTC from Asia, USA, and Europe has potential to impact AMR bacterial and seafood globally owing to continuous water mass movements assisted by ocean currents.

Our findings emphasise the need for urgent, coordinated national and international interventions to limit the use of antimicrobials, and limit the global spread of AMR. The proposed strategies are as follows: the reduction of waste including industrial runoff; correct waste disposal; reduction in discharge of chemicals from pharmaceutical plants; and the employment of effective technologies in hospital, urban, and industrial wastewater treatment. Furthermore, we suggest an appropriate modelling from Global Ocean Observing System to predict the OTC, heavy metals, and AMR bacteria distribution.

**Author Contributions:** Conceptualization, B.M., M.G. and F.T.; methodology, B.M. and M.G.; formal analysis, B.M. and M.G.; investigation, F.T.; resources, M.G. and F.T.; writing—original draft preparation, B.M., M.G. and F.T.; writing—review and editing, S.O.A., M.G., B.M., A.B. and J.C.; visualization, B.M., F.T. and A.B.; supervision, M.G. and F.T. All authors have read and agreed to the published version of the manuscript.

**Funding:** This research received no external funding.

**Institutional Review Board Statement:** Not applicable.

**Informed Consent Statement:** Not applicable.

**Data Availability Statement:** Not applicable.

**Conflicts of Interest:** The authors declare no conflict of interest.

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
