# Peer review of "Hospital Effluents and Wastewater Treatment Plants: A Source of Oxytetracycline and Antimicrobial-Resistant Bacteria in Seafood"

_sustainability, doi:10.3390/su132413967_

Round 1
Reviewer 1 Report
This work reviews the occurrence of oxytetracycline and ARB in hospital effluents and WWTPs. The topic of the manuscript is interesting and relevant data is reported. However, some concerns should be addressed.
- Authors describe the risks associated to the presence of oxytetracycline and antimicrobial resistant bacteria in water bodies, stating that current conventional technologies are not able to remove these pollutants. I recommend including information related to the technologies employed for hospital effluents or wastewater treatment. Recent reviews related to disinfection and pharmaceuticals removal from hospital wastewater can help the authors to complete the manuscript: Science of the Total Environment, 2021, 797, 149150. Environmental Pollution, 2021, 291,118233.
- Pharmaceuticals and ARB from hospital effluents come mainly from the urine since these chemical and biological pollutants are excreted by this route. This information should be detailed in the manuscript.
- I recommend increasing the literature search to today (2021).
- Results related to AMR bacteria is hospital wastewater (section 3.1.) only describes E. coli as microorganism. However, there exist other bacteria in hospital effluents that are more dangerous for human health. This information should be included. The following reference can help the authors: Journal of Cleaner Production, 2021, 320, 128865.
Author Response
We thank the expert referee for valuable feedback, insightful comments and for giving the time to review our work. Following their suggestions, we have included significant additional information and clarified key points. Our point-by-point response below addresses their specific comments. The response to review 1 are red. All the changes are in the manuscript and locations are indicated in the responses.

Reviewer 2 Report
The presented work is a review of data on the presence of oxytetracycline (OTC) and resistance (AMR) bacteria in hospital wastewater treatment plants and their distribution by ocean currents. The review is part of environmental research, which is currently of great importance. The authors collected available data from many countries. I have doubts about figures 1 and 2, which in my opinion should be removed because they do not enrich the work in any way.
Author Response
We thank the expert referees for valuable feedback, insightful comments and for giving the time to review our work. Following their suggestions, we have included significant additional information and clarified key points. Our point-by-point response below addresses their specific comments. The responds to review 2 are in blue.

Round 2
Reviewer 1 Report
The manuscript has been improved. All my questions have been correctly addressed, therefore, I recommend its publication